# Outcomes of Femoral Arterial Catheterisation in Neonates: A Retrospective Cohort Study

**DOI:** 10.3390/children9081259

**Published:** 2022-08-20

**Authors:** Lucy Turner, Vasiliki Alexopolou, Hanin Tawfik Mohammed Tawfik, Monica Silva, Charles William Yoxall

**Affiliations:** Neonatal Unit, Liverpool Women’s Hospital, Liverpool L8 7SS, UK

**Keywords:** femoral arterial catheter, neonate, intensive care, risk assessment

## Abstract

Background: To review the outcome of all femoral arterial catheter (FAC) insertions in a single, large neonatal unit over a 12 year period, we will describe the incidence of harms arising from FAC insertion and to identify risk factors associated with ischaemic injury. Methods: Retrospective survey of data relating to all episodes of FAC insertion in a single neonatal intensive care unit over a 12 year period up to 2020. Results: 146 FACs were inserted into 139 babies with a median (interquartile range) gestation and birth weight of 27 (24 to 37) weeks and 1092 (682 to 2870) g. Impaired limb perfusion occurred in 32 (22%). This was transient and recovered with no injury in 26 of the 32. There was an increased risk of impaired limb perfusion in babies with lower weight at the time of insertion; from 5.7% in babies over 3000 g to 34.7% in babies under 1000 g (relative risk 6.1 (1.5 to 24.6)). Six babies (4%) had ischaemic injury. Risk factors for ischaemic injury included weight below 1000 g (four cases), pre-existing partial arterial obstruction (two cases), concerns about limb perfusion prior to FAC insertion (two cases) and a delay in removing the FAC after recognition of the poor perfusion (five cases). Two clinicians inserted 71 (50%) FACs and had no associated injuries. Conclusions: FAC can be used in neonates, although there is a risk of ischaemic injury, particularly in very small babies. Our data can be used to inform decisions about patient selection for this procedure.

## 1. Introduction

Intravascular arterial access is used during neonatal intensive care for continuous accurate monitoring of arterial blood pressure, to measure arterial blood gases and to provide a reliable source of blood sampling. Commonly used arterial access sites are the umbilical artery catheter (UAC) and catheterisation of some peripheral arteries, such as the radial artery.

When neither the umbilical nor peripheral arteries are available, it is possible to establish arterial access using a femoral arterial catheter (FAC). This route is commonly used in adult and paediatric intensive care, but is less commonly used in neonatal care, where it is seen as a ‘last resort’ for establishing arterial access. A survey of practice across UK regional neonatal intensive care units in 2014 found that they were in use in 16 of 40 (41%) units [1]. Guidelines published online from several neonatal units in various countries also refer to the use of FACs in neonatal intensive care [2,3,4,5].

The leg has some protection against ischaemic injury during FAC insertion from collateral arteries [6], but limb ischaemia remains an associated risk with potentially life long implications. Concern about limb ischaemia was the most commonly reported reason why units did not use FAC in a previous survey [1].

There may also be concerns that the proximity of the FAC insertion site to the nappy of the area of the baby may also increase the risk of catheter related infection.

All intensive care procedures have associated risks and arterial catheterisation is no exception. Risks include catheter related bloodstream infection, haemorrhage and ischaemia or necrotic injury in the territory of the catheterised artery. These have been well described in both UAC and peripheral arterial artery catheterisation [7,8,9,10,11,12,13]. We reported the first description of FAC insertion using the Seldinger method in 2001 [14]. In that series of 21 FACs we found that in 4 babies (15%) the catheter had to be removed shortly after insertion due to ischaemia of the leg, but that in each case the ischaemia was transient with no necrotic injury. To our knowledge there have been no other reports that report about the outcomes of FAC only in neonates.

We have continued to use FACs in selected babies requiring arterial access in whom we have been unable to gain access using UAC or peripheral catheters since that time. In early 2020, 3 babies on our unit had ischaemic necrosis of the leg following FAC insertion. We have performed this study in response to those incidents and to update our understanding of the outcomes of FAC.

The aims of this study were to review the outcome of all FAC insertions in a single large neonatal unit over a 12 year period, to describe the incidence of harms arising from FAC insertion and to identify risk factors associated with ischaemic injury.

## 2. Materials and Methods

Babies who were deemed to require arterial access were those in whom greater precision in respiratory or circulatory monitoring was thought to be needed. These included babies receiving significant respiratory support in whom the greater precision of arterial blood gas measurements (compared to capillary blood gas measurements) was thought to be a better guide to therapy, or babies requiring very frequent blood gas measurements, in whom repeated capillary sampling was felt to be too numerous and distressing. Other eligible babies were those requiring circulatory support with inotropes in whom the precision afforded by continuous invasive blood pressure measurement was thought be a better guide to therapy than intermittent non-invasive measurement.

FAC insertion was performed in patients in whom arterial access was deemed to be necessary to facilitate care, as defined above, and in whom it had not proved possible to site a UAC, or a peripheral arterial catheter. The decision to site arterial access required an assessment of the benefits in the individual patient against the potential risks of the procedure. Although not specifically described in our unit policies, the threshold for femoral arterial catheter insertion was higher than the threshold for insertion of UAC or peripheral arterial catheters, with which we have greater experience.

All FACs were inserted by a consultant neonatologist or a senior trainee supervised by a consultant neonatologist. All catheters were inserted using the Seldinger technique over a wire introduced via a 22 or 24 gauge needle or cannula. The skin was cleaned with chlorhexidine solution prior to insertion (0.05% aqueous solution for babies < 26 weeks of age and <7 days of age, 0.5% solution in 70% alcohol in all other babies). The site was covered with an occlusive plastic sterile dressing. An infusion of 0.9% saline with heparin (1 unit/mL) was infused through all FACs at a rate of 1 ml/hour using an in-line bacterial filter.

This was a retrospective observational study. The electronic patient record (Badger 3 system and Badgernet full EPR, Clevermed, Edinburgh, UK) was used to identify all episodes of FAC insertion between 20 August 2008 and 11 May 2020. The start date was the date when catheter insertion data were routinely captured in our electronic patient record system and the end date was shortly after the third episode of ischaemic injury occurred in 2020, referred to above. Data extracted included: patient demographics, details of the catheter insertion procedure, duration of catheterisation, reasons for catheter removal, evidence of compromised limb circulation and the occurrence of ischaemic injury.

All positive blood cultures obtained from all subjects during the period when the FAC was in situ, and the 48 h following FAC removal, were collected from a clinical database in which they had been prospectively classified as Catheter Associated Blood Stream Infection (CABSI) or not. Definition of CASBI was made using an commonly used definition [15] (Table 1)

Each case of ischemic injury was further investigated to identify any other potential contributing factors. Further detailed analysis of babies with no injury was not performed.

Statistical analysis was performed using SPSS v26 (IBM, Portsmouth, UK). Mann-Whitney tests were used for continuous variables and Chi squared tests were used for categorical data. Our a priori clinical impression was that the risk of post insertion impaired limb perfusion was greater in smaller babies. For the purposes of analysis, the cohort was divided into approximate quartiles by weight using a heuristic approach to generate clinically recognisable groups (<1000 g, 1000 g to 1499 g, 1500 g to 3000 g and >3000 g). The risk of impaired perfusion was calculated for each group. The Relative Risk (RR) of impaired perfusion in each group compared to the reference group (those weighing > 3000 g) was calculated. The results refer to all catheters (with no allowance made for repeated catheters in 7 babies).

## 3. Results

146 FACs were inserted into 139 babies over the 12 year period. The demographics of the population and are shown in Table 2.

The side of insertion was recorded on 132 occasions. The FAC was inserted on the right side in 100 cases. Data relating to the type and size of catheter inserted was not of an analysable quality.

All patients were ventilated at the time of the insertion and 114/146 (78%) were also receiving inotropes. Catheters remained in-situ for a median (interquartile range) of 4.5 (1.9 to 9) days. A total of 937 days of invasive monitoring was provided to these babies by FAC.

The reason for catheter removal is shown in Table 3. 82 (56%) remained in situ until either no longer needed, the patient was transferred to paediatric intensive care or the patient’s death.

### 3.1. Bleeding

Significant bleeding was encountered in 2 patients. Both of these were found to have abnormal blood clotting. The catheter was removed in both cases and bleeding stopped with compression and treatment of the coagulopathy. Neither patient required blood transfusion.

### 3.2. Infection

12 FACs were recorded as being removed due to concern about infection. Positive blood cultures were obtained whilst an FAC was in situ, or within 48 h of its removal in 25 cases. Of these 9 were identifed as CABSIs. Definite pathogens were identifed in 6 cases; Pseudomonas aeruginosa in 2, Group B β haemolytic streptococcus (GBS) in 1, Candida species in 3 and Stenotropomonas Maltophilia in 1. Coagulase negative Staphylococcus (CONS) was identified, accompanied by at least 3 clinical features of infection, in the other 3 CABSI cases. The rate of CABSI was, therefore, 9 cases in 937 days of care or 9.6/1000 days of care.

4 of the positive blood cultures were obtained within 24 h of FAC insertion (1 case each of pseudomanas, GBS, Candida and CONS). It seems unlikely that the FAC was the source of the infection in these cases. Excluding these 4 cases gives a CABSI rate of 5.3/1000 days.

In all 9 cases classified as CABSI, the patient also had an indwelling central venous line (CVL) at the time of the blood culture, so it is not possible to know whether the source of the CABSI was the FAC or the CVL in any individual case.

Of the 16 positive blood cultures that were not classified as CABSI, 15 contained a pure growth of CONS and one contained a mixed growth of CONS with Micrococcus. None of these babies had 3 or more clinical features of infection. In 12 of these cases, there was an indwelling CVL, so even if these positive cultures were due to a CABSI, it would not be possible to determine whether the FAC or the CVL was the source of the infection in most cases.

### 3.3. Impaired Leg Perfusion

Concern about leg perfusion was cited as the reason for removal in 23 (16.4%). Concern about limb perfusion were also present in another 9 babies who had some other reason for removal recorded. Overall the number of FAC insertions causing impaired limb perfusion was 32 (21.8%).

There was no difference in the frequency of impaired perfusion by side of insertion or post natal age at the time of insertion (Table 3). Babies who experienced impaired perfusion had lower weights at the time of FAC insertion compared to babies who did not experience impaired perfusion; median (interquartile range) 990 g (732.5 to 1316.3) versus 930 g (1643 to 3200) *p* < 0.0001 (Table 4).

Table 4 shows the number, rate and relative risk of impaired limb perfusion by weight category. The proportion of insertions that caused impaired perfusion increased with decreasing body weight at the time of insertion from 5.7% in babies over 3000 g to 34.7% in babies under 1000 g. Compared to the rate in babies with a weight at insertion of >3000 g, the increased rate of impaired perfusion reached statistical significance in babies with weights between 1000 g to 1499 g (RR (95% confidence interval) 4.8 (1.1 to 28)) and <1000 g (RR (95% confidence interval) 6.1 (1.5 to 24.6)) (Figure 1, Table 5).

### 3.4. Ischaemic Injury

In 26 of the 32 episodes where there was concern about impaired perfusion, this was transient with normal perfusion of the leg immediately on removal of the FAC and no ischaemic injury. In 6 cases ischaemic injury occurred despite FAC removal. The rate of ischaemic injury was, therefore 4% of the total cohort.

There was no difference in the frequency of ischaemic injury perfusion by side of insertion or post natal age at the time of insertion (Table 5). Babies who experienced ischaemic injury had lower weights at the time of FAC insertion compared to babies who did not experience ischaemic injury; median (interquartile range) 755 g (672.5 to 1338.8) versus 1325 g (912.5 to 3022.5) *p* = 0.02 (Table 6).

Four of the 6 babies who experienced ischaemic injury had a weight under 1000 g at the time of insertion (Figure 1). Three babies developed total limb necrosis, 2 of these babies died of co-existing illnesses and one survived, but required limb amputation. Three babies developed ischaemic injury to one or more of their toes and all survived.

In two of the babies that sustained ischaemic injury there was some co-existing partial arterial obstruction (non-occlusive aortic thrombosis in one case and a co-existing, non-functioning, UAC which had not yet been removed in one case). In two other cases there had been concerns about impaired perfusion of the leg prior to catheter insertion. We do not have information about the presence of these factors in babies without injury, so cannot support our view that these factors influenced the outcome with any statistical analysis.

In five of the six cases of ischaemic injury there was a delay between the recognition of the impaired perfusion and the removal of the catheter of between 4 and 20 h. The median (interquartile range) interval between recognition of limb ischaemia and removal of the FAC in babies who experienced ischaemic injury was 5.3 (3.3 to 7.3) hours, compared to 3.3 (0.9 to 8.3) hours in babies with limb ischaemia who did not experience an ischaemic injury (*p* = 0.55).

### 3.5. Clinician Experience

FACs were inserted by 9 different clinicians during the study period, with a median (range) of 11 (3 to 55) insertions per clinician. Not all of the clinicians worked within the department during the whole of the period. The median (range) number of insertions per clinician by year of activity was 2.2 (0.3 to 4.7). 71 (50%) of the FACs were inserted by 2 clinicians, who each inserted more than 4 FACs per year of activity. None of the FACs inserted by these 2 clinicians was associated with an ischaemic injury.

## 4. Discussion

Our study is the largest published series of FAC insertion in the newborn and includes a significant number of very low birth weight babies. Although FAC insertion is practised in neonatal care, there is limited published information relating to the risks or benefits of the technique. Most published series combine results from neonatal and paediatric patients and report low complication rates [12].

FAC insertion allowed us to provide 937 days of invasive monitoring for critically ill babies on our unit during the 12 year period. Babies with a weight below 1000 g least benefit from FAC insertion as the rate of impaired perfusion was high and these catheters were frequently removed shortly after insertion.

FACs were inserted into the right femoral artery more commonly than the left; 100/132 (75.8%) versus 32/132 (24.2%). This is due to the fact that it is technically easier for a right handed operator to insert the FAC on the right side than the left. There is no evidence in our data to suggest that the side of insertion has any impact on the risk of circulatory disturbance.

It was difficult to be precise about the rate of infection due to FAC. By a conventional definition, the rate of CABSI was between 5.3 and 9.6/1000 days of care. All of these babies also had CVL in place, so it is not possible to know how many, if any, of these infections were due to the FAC. The calculated rate appears to be within the reported rates of CABSI reported in other neonatal populations [16].

In our series, 15% (15/97) of babies with a weight above 1000 g had a failed procedure, with removal of the catheter shortly after insertion due to impaired limb perfusion, but the rate of ischaemic injury in this group was only 2% (2/97). Overall, outcomes were good in babies with a weight above 1000 g at the time of insertion, with no pre-existing evidence of lower limb arterial vulnerability, and who had catheters removed as soon as impaired perfusion was recognised.

Ischaemic injuries did occur in our cohort and there is clearly a risk of causing injury with FAC insertion. The overall rate of ischaemic injury in our cohort was 4%.

In 4 of the 6 babies who sustained an injury in our cohort, the weight at insertion was below 1000 g. There were also other pre-existing reasons to suspect that limb perfusion could be compromised by FAC insertion that were not appreciated prior to the insertion in 4 of these 6 babies. In 5 of the 6 babies with ischaemic injury, there was a delay between the recognition of the poor limb perfusion and FAC removal. Our data do not show that the duration of delay between the recognition of the ischaemia and the removal of the FAC is greater in the babies who subsequently experienced ischaemic injury compared to the duration of delay in those who did not experience an injury, but the number of injuries in the cohort is low, so this could be a type 1 error.

This experience is based on observations at a single centre and we have no information about the rates of complication seen in other centres, so generalisability of these results is not assured.

The decision to site any arterial catheter in a baby undergoing intensive care should always be made after careful consideration of the likely risks and benefits for that individual patient. In many babies adequate monitoring can be provided using less invasive techniques for monitoring (for example; non-invasive blood pressure monitoring, functional echocardiography, pulse oximetry, end tidal capnography, capillary blood gas measurement). Given the margin of error associated with each of these techniques however, it may be beneficial to use an arterial catheter for monitoring in the sickest babies, with the highest requirement of respiratory or circulatory support.

Other forms of arterial catheter are also known to have an associated risk of ischaemic injury. There are widely varying estimates of the incidence of arterial thrombosis occurring in babies following UAC insertion, which is probably a reflection of the populations studied and diagnostic criteria applied in each published study;

A recent systematic review of UAC related adverse events reported a pooled estimate for UAC related thrombosis diagnosed by radiography or by clinical signs of 8.16% [10]A meta analysis of trials comparing high UAC position compared to low UAC position reported the results of 5 studies including 1587 subjects, with “clinical vascular compromise” reported to occur in 394 subjects (24.8%). [11]A study of sequential ultrasound examination of 75 UACs found evidence of catheter related thrombus in 9 cases (12%). [12].

Ramesethu reviewed the published rate of ischaemic injury in babies who had radial or ulnar arterial lines inserted and reported a rate of 5% [9].

Although the rate of impaired perfusion following FAC insertion in our series was 22%, the incidence of ischaemic injury in our series was only 4%. This rate appears to be comparable or better than the rates of ischaemic injury reported in the literature following other forms of arterial catheterization in the newborn. It is difficult to be certain about this observation because of the differences in diagnostic criteria between these various studies and differences in the populations under study. In our cohort FAC was only used in the sickest babies, in whom other forms of arterial access was not possible. It is possible that the risks were higher in our population than in the studies in which UAC or peripheral arterial line were inserted. None of the studies reporting on the rates of vascular injury following UAC or peripheral arterial catheter insertion have reported the incidence in relation to weight at the time of insertion. Further investigations on the outcomes of UAC and peripheral arterial catheter insertion in a similar population to that studied in our series, using similar, clinical, diagnostic criteria, would allow a comparison of the relative risks of FAC insertion compared to other approaches.

If FAC insertion is being considered in a baby it is clearly important that there is careful patient selection. In babies with a weight below 1000 g the risks may outweigh the benefits as 35% of our series required removal immediately because of impaired perfusion, and 25% of these sustained an ischaemic injury. We suggest that FACs should not be inserted in babies in whom there are already concerns about impaired leg perfusion and in babies weighing <1000 g the risk assessment is likely to be against insertion unless a very strong case of monitoring need can be made. Careful observation of subsequent limb perfusion should occur, with prompt removal of any FAC causing impaired limb perfusion.

The need for FAC insertion is uncommon so the exposure of any individual professional to this procedure is limited. Our data, however, suggests that serious injury is much less likely when FACs are inserted by individuals with the most experience. Given this, we suggest that the decision to insert these catheters, or to provide any other uncommonly used high risk procedure should involve a discussion between at least 2 clinicians.

## 5. Conclusions

We conclude that FAC is potentially useful as a rescue procedure to establish arterial access in carefully selected neonates but requires careful patient selection with intensive monitoring and prompt removal if impaired limb perfusion is seen.

## Figures and Tables

**Figure 1 children-09-01259-f001:**
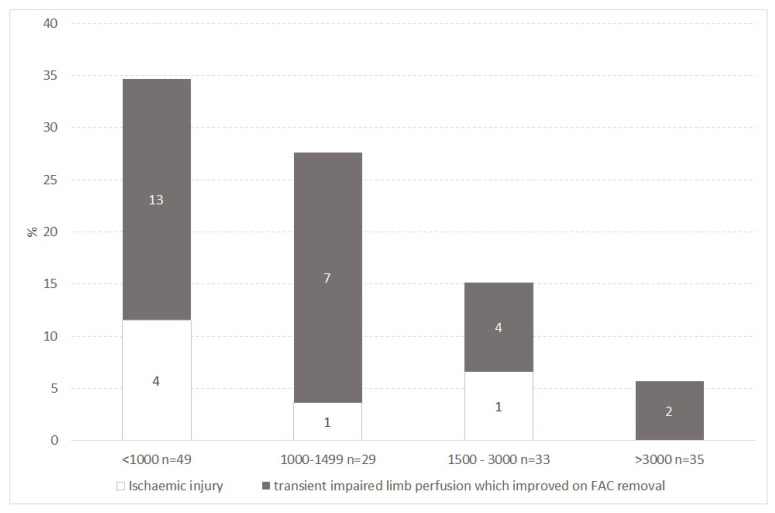
Proportion of FAC causing transient impaired limb perfusion which improved on FAC removal and ischaemic injury by weight category at time of insertion. n = the number of FAC inserted in each weight category.

**Table 1 children-09-01259-t001:** Case definition criteria for Catheter Associated Blood Stream Infection (CABSI).

Criteria	
1	Positive Blood culture taken whilst intravascular catheter is in place or within 48 h of its removal
2	Pure growth of a definite pathogen
3	Mixed growth, or growth of a skin commensal
4	Clinical signs of infection (a) Tachypnoea or clinically relevant increase in oxygen requirement or ventilatory support (b) Hypotension (c) Hypoglycaemia or Hyperglycaemia (d) Impaired peripheral perfusion (e) Lethargy/irritability/poor handling (f) Temperature instability (g) Ileus/feed intolerance (h) Fall in urine output (i) Metabolic acidosis (Base excess < −10)

CABSI = Criteria 1 + 2 or Criteria 1 + 3 + at least 3 clinical signs of infection from Criteria 4.

**Table 2 children-09-01259-t002:** Demographics and age of insertion for the whole cohort.

	Median	Lower Quartile	Upper Quartile
BWt (g)	1012.5	682	2870
Gestation (weeks)	27	24	37
Wt at insertion (g)	1298	882	3000
Age at insertion (days)	12.03	2.02	24.39

**Table 3 children-09-01259-t003:** Reasons for FAC removal.

Reason for Catheter Removal	Number	Percentage
No longer needed	44	30.1
Still present when baby transferred out	10	6.8
Still present at time of death	28	19.2
Fell out	5	3.4
Occluded	19	13.0
Concern about infection	12	8.2
Haemorrhage	2	1.4
Concern about perfusion	23	16.4
Other	3	2.1

**Table 4 children-09-01259-t004:** Weight at time of insertion, post natal age and side of insertion in babies with and without impaired perfusion following FAC insertion.

	Impaired Perfusion	No Impaired Perfusion	*p*
Weight at time of insertion (g) ^#^	990 (732.5 to 1316.3)	930 (1643 to 3200)	<0.0001
Right sided insertion	22/32	78/100	0.29
Post natal age at time of insertion ^#^	16.5 (3.8 to 23.8)	10.1 (1.8 to 24.8)	0.26

# Data shown as Median and interquartile range.

**Table 5 children-09-01259-t005:** Rate of impaired perfusion by weight category at time of insertion. RR = relative risk.

Weight at Time of Insertion (g)	n	Impaired Perfusion	No Impaired Perfusion	RR	95% CI
<1000 *n* = 49	49	17	32	6.1	1.5 to 24.6
1000–1499 *n* = 29	29	8	21	4.8	1.1 to 21.0
1500–3000 *n* = 33	33	5	28	2.6	0.5 to 12.7
>3000 *n* = 35	35	2	33	Comparison group	

**Table 6 children-09-01259-t006:** Weight at time of insertion, post natal age and side of insertion in babies with and without ischaemic injury following FAC insertion.

	Ischaemic Injury	No Ischaemic Injury	*p*
Weight at time of insertion (g) ^#^	755 (672.5 to 1138.8)	1325 (912.5 to 3022.5)	0.02
Right sided insertion	5/6	95/136	0.5
Post natal age at time of Insertion (days) ^#^	5.75 (14.4 to 26.1)	11.8 (1.8 to 24.8)	0.67

# Data shown as Median and interquartile range.

## Data Availability

The data presented in this study are available on request from the corresponding author.

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
