# Peer review of "Outcomes of Femoral Arterial Catheterisation in Neonates: A Retrospective Cohort Study"

_children, 2022, doi:10.3390/children9081259_

Round 1

Reviewer 1 Report

Please the Authors to refer to the attached file

Author Response

I have attached my reply as a separate file

Reviewer 2 Report

This thoughtful and informative paper by Turner, et al makes important contributions to our understanding of the outcomes of femoral arterial catheterization in neonates. The paper is well written and novel in its conclusions. I have the following minor suggestion for the authors: 

Page 5- Discussion line 166: You have mentioned that the outcomes were good for FAC in babies >1000 gm weight at the time of insertion. The rate of ischemia in these babies was approximately 15% (15 out of 97), which is still high, especially in the group with babies with a weight of 1-1.5 kg. Would you be able to rephrase it to something like ‘babies with weight >1000g had better outcomes after FAC insertion compared to babies <1kg’? 

Author Response

(The authors gave the same response as above.)

Reviewer 3 Report

This is a good manuscript on a really interesting topic. I will point out the issues that bother me:

-         I am very reticent of the articles/manuscripts that use the word “babies”. I would recommend the authors to replace it with “infants”

-        The details of the insertion procedure were extracted from the database – what are those specific details? Could they be relevant to the study?

-        The authors should take another look in the statistics to see whether the postnatal age at insertion of the FAC has something to do with the adverse events, namely ischemic injuries – could there be a statistical correlation between the early age at insertion and the appearance of ischemia?

-        On row 165, remove “eg” from the squares

-        Please remove the last paragraph from the Discussion section and place it to the Conclusions section

Author Response

(The authors gave the same response as above.)

Reviewer 4 Report

Dear Authors

That is great that You concern in the topic of FAC

I would suggest to avoid using abrevation in the abstract. Could You do it?

What do You think about it?

Could You change it?

Author Response

(The authors gave the same response as above.)
